# Feasibility and Implementation of an Oncology Rehabilitation Triage Clinic: Assessing Rehabilitation, Exercise Need, and Triage Pathways within the Alberta Cancer Exercise–Neuro-Oncology Study

Lauren C. Capozzi [1,2,*], Julia T. Daun [1], George J. Francis [2,3], Marie de Guzman Wilding [4], Gloria Roldan Urgoiti [3], David Langelier [5,6] and Nicole Culos-Reed [1,3,4]

1   Faculty of Kinesiology, University of Calgary, Calgary, AB T2N 1N4, Canada; jtdaun@ucalgary.ca (J.T.D.); nculosre@ucalgary.ca (N.C.-R.)
2   Department of Clinical Neurosciences, Cumming School of Medicine, University of Calgary, Calgary, AB T2N 1N4, Canada; george.francis@ucalgary.ca
3   Department of Oncology, Cumming School of Medicine, University of Calgary, Calgary, AB T2N 1N4, Canada; gloria.roldanurgoiti@albertahealthservices.ca
4   Supportive Care: Psychosocial and Rehabilitation Oncology, Cancer Care Alberta, Alberta Health Services, Calgary, AB T2S 3C3, Canada; marie.deguzmanwilding@albertahealthservices.ca
5   Department of Medicine, Division of Physical Medicine and Rehabilitation, University of Toronto, Toronto, ON M5S 1A8, Canada; david.langelier@uhn.ca
6   Department of Supportive Care, Cancer Rehabilitation and Survivorship, Princess Margaret Hospital, Toronto, ON M5G 2C1, Canada
*   Correspondence: lcapozzi@ucalgary.ca

**Abstract:** Individuals living with and beyond cancer face physical impairment and inactivity in survivorship. Neuro-oncology populations have especially high rates of sedentary behaviour and functional deficits, including impaired balance, motor skills, and cognition. Our purpose was to assess the rehabilitation and exercise needs of patients with brain tumours while examining the feasibility of a rehabilitation triage clinic as a part of the Alberta Cancer Exercise–Neuro-Oncology study, where patients were referred to a triage clinic, where health, neurologic, and functional status was assessed, followed by a referral to one or multiple resources, including exercise, physiotherapy, occupational therapy, or physiatry. Qualitative perception of the triage clinic was collected. Overall, the triage clinic was feasible and safe for participants, facilitating referral into rehabilitation and exercise resources. Pre-determined enrollment and attendance rates were met, but referral rates to the triage clinic were not met. Oncology clinic staff reported forgetting to refer patients or uncertainty of who was appropriate for rehabilitation as barriers. Oncology clinic-based screening may improve the identification of patients who are sedentary or have a physical impairment. A proposed screening tool, the Cancer Rehabilitation and Exercise Screening Tool (CREST), is presented within our Cancer Rehabilitation and Exercise Pathways Model. The CREST can identify patients who are sedentary or have a functional impairment, facilitating referral to appropriate rehabilitation resources and ultimately improving patient recovery and functioning.

**Keywords:** brain tumour; cancer rehabilitation; exercise; feasibility; screening for impairment

## 1. Introduction

Due to improved screening and treatment, death rates for all cancer types combined have decreased by 33% since 1991 [1]. With increased survival, those living with and beyond cancer face an increased burden of physical and functional morbidity as well as diminished psychosocial well-being, resulting in lower quality of life into survivorship [2–4].

To address physical and functional impairment following a cancer diagnosis, multidisciplinary rehabilitation and exercise programs have been developed. Specifically, cancer

physiatry (physical medicine and rehabilitation physicians with a specialty in oncology), physiotherapy, occupational therapy, speech–language therapy, lymphedema management, as well as exercise prescription and counselling have strong evidence to support their important role in the care of cancer patients throughout the cancer journey [5–7].

Individually, these cancer rehabilitation and exercise interventions have been shown to improve function, psychosocial well-being, and survival [6,8–10]. Unfortunately, widespread access to cancer rehabilitation and exercise resources for individuals living with and beyond cancer lags behind those organized for patients with other chronic conditions, such as heart disease, for which rehabilitation and exercise are part of standard care [11]. There is thus an "evidence to practice" gap, with system-wide access to rehabilitation and exercise programs clinically lacking in many high-quality oncology care systems [6]. Previous reports comment on the essential component of rehabilitation and exercise in comprehensive cancer care [12,13]. Despite this, the development of cancer rehabilitation and exercise programs within clinical oncology care settings has been delayed, in part due to the lack of a specific implementation plan with effective patient screening, triage, and referral pathways [6,14,15].

To improve patient access to rehabilitation and exercise resources, clinical implementation to optimize patients receiving the right rehabilitation and exercise care at the right time must include: (1) screening patients for impairments and inactivity, (2) the development of triage resources to help with decision making for appropriate exercise and rehabilitation services, (3) sustainable system-embedded referral pathways [15,16], and (4) additional evidence-based rehabilitation and exercise programs to serve patients. Currently in Canada, there is a lack of system-embedded screening and triage tools, as well as referral pathways for cancer rehabilitation and exercise. Many programs rely on oncologists or nursing staff to identify patients in need of services, which previous research has shown falls short for patients. For example, Cheville and colleagues surveyed patients on 27 cancer-related symptoms, signs, and functional problems, and also reviewed electronic medical records (EMR) for oncology documentation [17]. They found a total of 65% of patients reported a functional impairment amenable to rehabilitation, yet only 6% of these problems were reported in the EMR by oncologists. Non-functional symptoms, including pain, weight loss, and nausea, however, were reported 49% of the time. This may be due to a lack of time, a lack of specific training to screen for functional impairment, or a lack of knowledge of rehabilitation and exercise resources. This disparity reinforces the need for standardized screening for all patients to effectively identify those with functional impairment, and implemented clinical pathways that can facilitate triage and referral to appropriate resources. The screening, triage, and referral approach is supported by extensive work in the area of psychosocial oncology, where effective screening for distress can improve the identification of affected patients, allowing for referral to appropriate services and leading to significantly decreased levels of distress when compared to not screening [18–20]. Applying this same principle to functional impairment and inactivity has the potential to significantly improve patient care and survivorship. Multiple call-to-action statements agree with the need for improved and integrated screening, triage, and referral pathways, and note that more research is needed in this important area [15,16].

Following the identification of patients with functional deficits or concerns through screening, it becomes essential to establish triage and referral pathways. In most cancer care systems in Canada, these are not well established for both rehabilitation and exercise. Santa Mina and colleagues [21] proposed a physical activity referral pathway, which was recently expanded upon by Wagoner et al. [22] as an example of triage pathways to rehabilitation and exercise. These models provide a clinical framework and are currently being studied and implemented [23,24]. Additionally, Covington and colleagues have proposed the Exercise in Cancer Evaluation and Decision Support (EXCEEDS) algorithm that is currently being studied, and have encouraged researchers to evaluate their evidence-based clinical decision-making referral tool in a variety of tumour groups [14].

Therefore, the objective of this research was to identify rehabilitation and exercise needs in an underserved oncology population, and study triage and referral processes to enhance patient rehabilitation and care. This manuscript presents data on the feasibility of the rehabilitation and exercise triage clinic as part of the Alberta Cancer Exercise–Neuro-Oncology study (ACE-Neuro) [25]. Specifically, the implementation of the triage clinic is reported, including the (a) assessment of rehabilitation and exercise needs of patients with brain tumours (i.e., neuro-oncology patients) and (b) the triage and referral of participants to physiatry, physiotherapy, occupational therapy, and/or exercise (i.e., ACE-Neuro) based on pre-determined cut-offs. Neuro-oncology patients were selected as a population of interest as they face unique functional challenges given side effects related to tumour location and treatments. They frequently experience cognitive, physical, and psychological impairments, and often report that their needs are not adequately addressed [26,27]. Unfortunately, methods to effectively screen and refer neuro-oncology patients to appropriate rehabilitation interventions are lacking [26,27]. Fortunately, there is great potential to continue to expand on the rehabilitation and exercise evidence for patients with brain tumours, including effective methods to identify patients in need and refer them to tailored rehabilitation programs [5,12].

Ultimately, the purpose of this work is to improve the identification of functional impairment and inactivity among patients with brain tumours, and identify effective strategies for triage and referral to appropriate rehabilitation and exercise resources. Our hope is that this will help to establish efficient pathways in rehabilitation oncology, so all cancer patients can be screened and receive appropriate rehabilitation and exercise care at the right time.

## 2. Materials and Methods

### 2.1. Study Design and Ethical Approval

This study was approved by the University of Calgary Health Research Ethics Board of Alberta (HREBA)–Cancer Committee (CC)–HREBA.CC-20-0322, and is a component of a larger study, Alberta Cancer Exercise–Neuro (ACE-Neuro) [25]. The triage clinic was conducted in Calgary, Alberta, and does not include ACE-Neuro patients from the Edmonton, Alberta site. This study was a mixed-methods descriptive study reporting on feasibility outcomes.

### 2.2. Study Outcomes

Feasibility was the primary outcome with both quantitative and qualitative components. Quantitatively, feasibility was defined a priori as a referral rate of at least 50%, an enrollment rate of at least 50%, and a triage clinic attendance rate of at least 60%. These feasibility thresholds were based on other feasibility work in exercise oncology as well as on feedback from the clinical team [28–30]. Specifically, given the poor survival prognoses and high symptom burden of neuro-oncology patients, lower criteria were expected.

Referral rate was defined as the number of patients referred from the clinical team to the ACE-Neuro out of the total number of patients seen in the clinic over the recruitment period (i.e., from 16 April 2021 to 2 December 2022). The enrollment rate was defined as the number of patients who enrolled after hearing the full study description out of the total number of patients referred. Finally, the triage clinic attendance rate was defined as the number of people who attended the triage clinic assessment out of the total number enrolled. Feasibility was also assessed by examining the safety of the triage clinic and documenting any adverse events. Adverse events were tracked using a standardized adverse event reporting system that classifies adverse events as level 1 (minor incident with no lost time beyond the day of injury; temporary, immediate care), level 2 (medical aid with no lost time beyond the day of injury; medical care beyond first aid), or level 3 (serious injury or death) [25]. Feasibility and acceptability were also assessed qualitatively via semi-structured interviews with participants.

### 2.3. Participants

All neuro-oncology patients with a primary brain tumour (benign or malignant) over 18 years of age and able to consent in English were eligible to participate in the study. Patients with secondary brain metastases were excluded. Participants could be at any stage in the treatment pathway (pre, on, or post-treatment). Participants were recruited at the Tom Baker Cancer Centre in Calgary, Alberta, Canada. As the primary outcome was feasibility, no a priori sample size was calculated.

Eligible neuro-oncology patients were approached by the study team after obtaining consent to contact. If interested, a clinical team member (nurse or oncologist) sent a referral to the Rehabilitation Oncology department via the electronic medical system [31]. Patients were also able to self-refer to the study via a study brochure or poster located within relevant clinic areas. Once referred, the study coordinator contacted the patient to review study eligibility and details and obtain consent to participate. Patients who agreed to participate did so via REDCap, a secure web application (Research Electronic Data Capture; REDCap) [25]. After providing informed consent, participants completed the health and medical history screening, including a Health History Questionnaire (i.e., to collect medical history) and Identifying Information Questionnaire (i.e., to collect demographics), as well as the Physical Activity Readiness Questionnaire, PAR-Q+. All screening was completed via REDCap [25]. Once consent and initial questionnaires were completed, the ACE-Neuro study coordinator (JTD; clinical exercise physiologist) reviewed participant health histories via chart review and phone call, and participants were booked into the triage clinic.

### 2.4. Triage Clinic

The triage clinic was led by a physical medicine and rehabilitation resident physician (LCC) and the ACE-Neuro study coordinator (JTD; clinical exercise physiologist). Participants were booked for a 45-min appointment, during which their medical and functional histories were reviewed, and a full central and peripheral neurological examination and the Short Physical Performance Battery Protocol (SPPB) were performed [32]. From this, the Karnofsky Performance Status (KPS) and Eastern Cooperative Oncology Group (ECOG) scores were determined. Criteria for triage included the SPPB, ECOG, and KPS, as well as previously published referral recommendations from Covington and colleagues [14] and pre-determined cut-offs from our clinical team. These pre-determined cut-offs were developed following consultation and deliberation with a multidisciplinary team, including rehabilitation clinical team leaders, physiotherapists, occupational therapists, physical medicine and rehabilitation doctors, behavioural medicine researchers, and clinical exercise physiologists. Please see Figure 1 for the triage clinic criteria. After the assessment, participants were then triaged and referred to the ACE-Neuro exercise study, Cancer Physiatry, Rehabilitation Oncology (i.e., Physiotherapy/Occupational Therapy), or a combination of these services. As part of the ACE-Neuro study, if triage to the ACE-Neuro exercise study was not appropriate after the triage clinic assessments, patients could be re-assessed in the triage clinic once deemed appropriate by their clinical team (i.e., oncologist, physiotherapist, or occupational therapist).

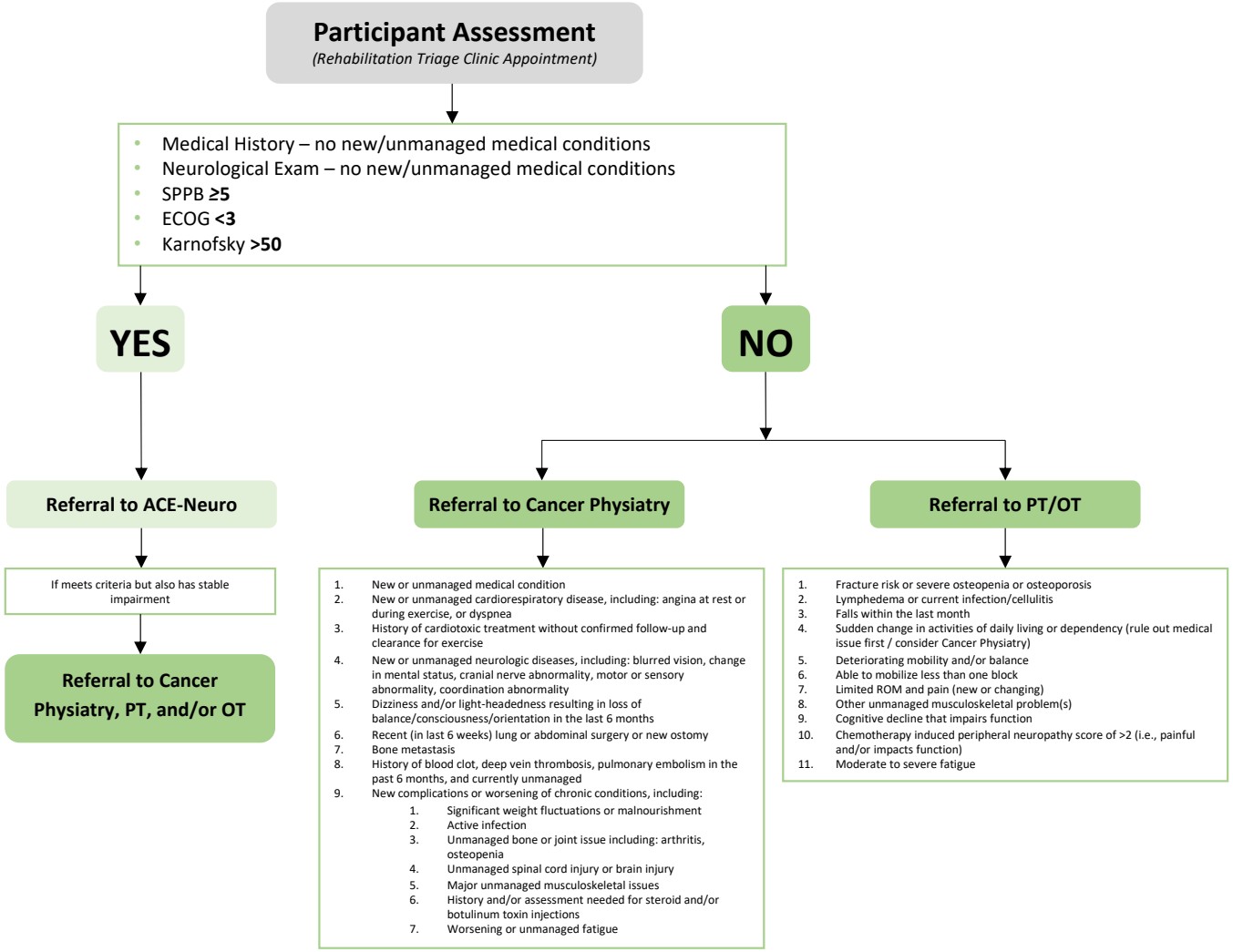

Adapted from Covington et al. 2021 and Campbell et al. 2019

*SPPB – Short Physical Performance Battery Protocol score; ECOG – Eastern Cooperative Oncology Group score; KPS – Karnofsky Performance Status; PT – physiotherapy; OT – occupational therapy*

**Figure 1.** Rehabilitation triage clinic criteria [12,14].

### 2.5. Study Measures

#### 2.5.1. Identifying Information Questionnaire and Health History Questionnaire

Both demographic and medical history were collected via patient report and chart review. Demographic history included participants' self-reported age, sex, self-identified gender, self-identified ethnicity, education, annual family income, and marital and employment status. Medical history included type of primary brain tumour, stage, treatment status, treatment types received, smoking status, alcohol intake, medical co-morbidities, and cancer-related co-morbidities. Participants also completed the Physical Activity Readiness Questionnaire (i.e., PAR-Q).

#### 2.5.2. Health-Related Fitness Measures

Health-related fitness measures included height and weight, resting heart rate, and blood pressure. Body mass index was calculated using height and weight.

### 2.5.3. Short Physical Performance Battery (SPPB)

The SPPB consists of a group of three tests examining gait speed, chair stand speed, and balance testing [33]. It is a validated tool used to predict risk for mortality, nursing home admission, and disability [33]. It is scored from 0 (worst performance) to 12 (best performance). A score of 5 or higher was necessary for direct referral to the ACE-Neuro exercise study. See Figure 2 for a summary of the SPPB.

## The Short Performance Physical Battery (SPPB) Protocol

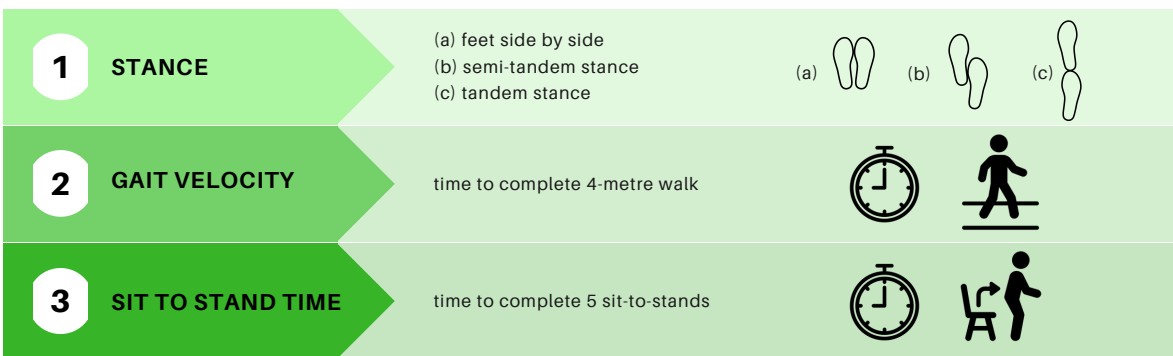

**Figure 2.** Short physical performance battery protocol summary.

### 2.5.4. Neurological Examination

A neurological examination was performed by the resident physician, consisting of a cognitive screening assessment as well as a physical examination. Cognitive screening consisted of examination of orientation, registration, recall, and language (speaking, reading, and writing). A cranial nerve screening examination was conducted, followed by a motor examination for tone, reflexes, bulk, and power. Finally, a sensory examination for light touch and pinprick sensation was conducted, and coordination was tested.

### 2.5.5. Karnofsky Performance Status (KPS)

KPS is a validated assessment tool for functional impairment, ranging from 100 (normal, no complaints, no sign of disease) to 0 (death). Each increment has well-defined criteria, which were used to classify study participants following a review of their health history and physical examination [34]. A score of 50 or higher was necessary for direct referral to ACE-Neuro.

### 2.5.6. Eastern Cooperative Oncology Group score (ECOG)

The ECOG is a validated assessment tool to assess functional status, scored from 0 (fully active, able to carry on all pre-disease performance without restriction) to 5 (death). As with the KPS, each increment has well-defined criteria used to classify study participants following their health history review and physical examination [35]. A score of 3 or lower qualified participants for the ACE-Neuro exercise study.

### 2.6. Qualitative Interviews

To obtain participant perspectives on triage clinic safety, acceptability, and satisfaction, semi-structured interviews were conducted with participants and members of the clinical team (i.e., oncologists, nurses, and administrators). We sampled and invited participants to a 15- to 30-min interview with the ACE-Neuro study coordinator (JTD) at the location of their choosing (i.e., via Zoom or in-person) at various times across the study duration. Specifically, participants were interviewed during or after the ACE-Neuro 12-week exercise intervention, and members of the clinical team were interviewed at various time points during the study recruitment period, with the aim of gathering varied perspectives to inform the clinical integration of processes specifically. Interviews were recorded by end-

to-end encrypted Zoom (online) or with an audio recording device (in-person). Examples of questions asked during the interview are presented in Appendix C.

*2.7. Statistical Analysis*

2.7.1. Quantitative Data

Descriptive characteristics of participants are presented using mean ± standard deviation or percentages. Feasibility was reported using percentages related to the predetermined thresholds mentioned above. Descriptive results, using mean ± standard deviation or percentages, are also reported for the SSPB, KPS, ECOG, and neurological examination results.

2.7.2. Qualitative Data

Interviews were transcribed verbatim via ExpressScribe [36], managed in NVivo 12 [37], and analyzed by one author (JTD) using conventional content analysis [38]. This iterative process included reading the transcripts, coding the data, and generating category descriptions. To ensure a rigorous process, a reflexivity journal was kept by JTD, and critical review and discussion with two other authors (LCC and SNC-R) occurred across the study process [39]. To enhance readability of participant quotes, repetitive words, identifiable information, and mumbled speech were replaced with brackets: [ . . . ].

## 3. Results

*3.1. Demographics and Feasibility*

See Table 1 for participant demographics and Table 2 for participant clinical characteristics and health history. The average age of participants was 51 ± 13.5, and the average time since diagnosis was 78.2 ± 101.7 months. The most commonly diagnosed brain tumour was glioblastoma (*n* = 19). Please see Appendix A for participant co-morbidities and cancer-related side effects.

**Table 1.** Participant demographics, *n* = 54.

| Demographic Variable | No. of Patients |
|---|---|
| Sex | |
| Male | 25 |
| Female | 29 |
| Self-Identified Gender | |
| Male | 25 |
| Female | 29 |
| Age: Mean ± SD, years | 51 ± 13.5 (range: 29–81) |
| Marital Status | |
| Never Married | 4 |
| Married | 43 |
| Common Law | 1 |
| Separated | 1 |
| Divorced | 5 |
| Education | |
| Some High School | 3 |
| Completed High School | 4 |
| Some University/College | 5 |
| Completed University/College | 28 |
| Some Graduate School | 3 |
| Completed Graduate School | 11 |

**Table 1.** *Cont.*

| Demographic Variable | No. of Patients |
|---|---|
| Annual Family Income, CDN$ | |
| <$20,000 | 3 |
| $20,000–$39,999 | 6 |
| $40,000–$59,999 | 7 |
| $60,000–$79,999 | 6 |
| $80,000–$99,999 | 2 |
| >$100,000 | 14 |
| Prefer not to answer | 16 |
| Employment Status | |
| Short-Term Disability | 1 |
| Long-Term Disability | 21 |
| Retired | 17 |
| Part-Time | 2 |
| Homemaker | 3 |
| Full-Time | 5 |
| Unemployed | 2 |
| Other | 3 |
| Self-Identified Ethnic Origin or Ancestry | |
| British | 9 |
| Western European | 12 |
| Eastern European | 3 |
| Northern European | 4 |
| Southern European | 1 |
| Eastern and Southern Asia | 6 |
| African | 2 |
| Other | |
| Canadian | 3 |
| Australian | 1 |

**Table 2.** Clinical characteristics and participant health history, *n* = 54.

| Clinical Characteristic | |
|---|---|
| Time Since Diagnosis: Mean ± SD, months | 78.2 ± 101.7 |
| Type of Primary Brain Tumour | Number of Participants |
| Glioblastoma | 19 |
| Oligodendroglioma | 16 |
| Astrocytoma | 12 |
| Meningioma | 3 |
| Medulloblastoma | 1 |
| Presumed Glioma | 1 |
| Germinoma | 1 |
| Malignant Glioma Not Otherwise Specified | 1 |
| Histologic Grade | |
| I | 2 |
| II | 9 |
| III | 14 |
| IV | 22 |
| Unknown | 7 |
| Treatment Status | |
| Pre-Treatment | 1 |
| On Treatment | 14 |
| Off Treatment | 32 |

**Table 2.** *Cont.*

| Clinical Characteristic | |
|---|---|
| Treatment Type | |
| Surgery Alone | 1 |
| Surgery + Chemoradiation + Adjuvant Chemotherapy | 18 |
| Surgery + Radiation | 3 |
| Surgery + Chemoradiation | 30 |
| Surgery + Chemotherapy | 1 |
| Chemoradiotherapy | 1 |
| Smoking Status | |
| Never Smoked | 36 |
| Ex-Smoker | 16 |
| Occasional Smoker | 1 |
| Regular Smoker | 1 |
| Alcohol Drinking Status | |
| Never Drinker | 10 |
| Ex-Drinker | 12 |
| Occasional Drinker | 21 |
| Social Drinker | 9 |
| Regular Drinker | 2 |

Note: $n$ = 54 participants, $n$ = 1 re-referral, seen in clinic twice.

Figure 3 presents the study flow chart. Recruitment was open for 20 months between April 2021 and December 2022. On average, 14 newly diagnosed neuro-oncology patients were seen at the Tom Baker Cancer Centre neuro-oncology clinic per month (a total of 280 patients were seen during the recruitment period). Of those, 86 were referred by a clinician to the triage clinic (referral rate of 31%). Approximately 194 patients were not referred due to (1) the clinical team forgetting to refer, (2) patient lack of interest, and (3) clinical judgment (e.g., a patient requiring palliative care, a patient unable to understand/speak English, or the clinical team being unsure of patient's rehabilitation needs). In addition to patients referred from the neuro-oncology clinic, 10 self-referred to the study, for a total of 96 patients being referred to the study. Of the 96 referred patients, 93 met the eligibility criteria. Three patients were excluded due to not being diagnosed with a primary brain tumour ($n$ = 1), unable to consent in English ($n$ = 1), or being diagnosed under the age of 18 ($n$ = 1). Of the 93 eligible, 57 enrolled in the study and completed informed consent (enrollment rate of 61%). Of the 36 patients who did not enroll, 15 were not interested, 12 were unable to be contacted, 8 had disease progression, and 1 moved to another country. Of the 57 enrolled participants, 54 attended the triage clinic (attendance rate of 94.7%). Reasons for non-attendance included time constraints ($n$ = 2) and not being interested at this time ($n$ = 1). One patient was seen in the triage clinic twice. On this patient's first visit to the triage clinic, they did not meet the ACE-Neuro exercise inclusion criteria and were referred to physiotherapy to improve physical function. They were later re-referred to the triage clinic, re-assessed, and triaged to exercise. The total number of participant assessments is thus $n$ = 55. No adverse events occurred during the triage clinic. The average time from referral to initial contact was 10.3 ± 8.9 business days, and the average time to triage clinic visit was 22.2 ± 20.0 business days.

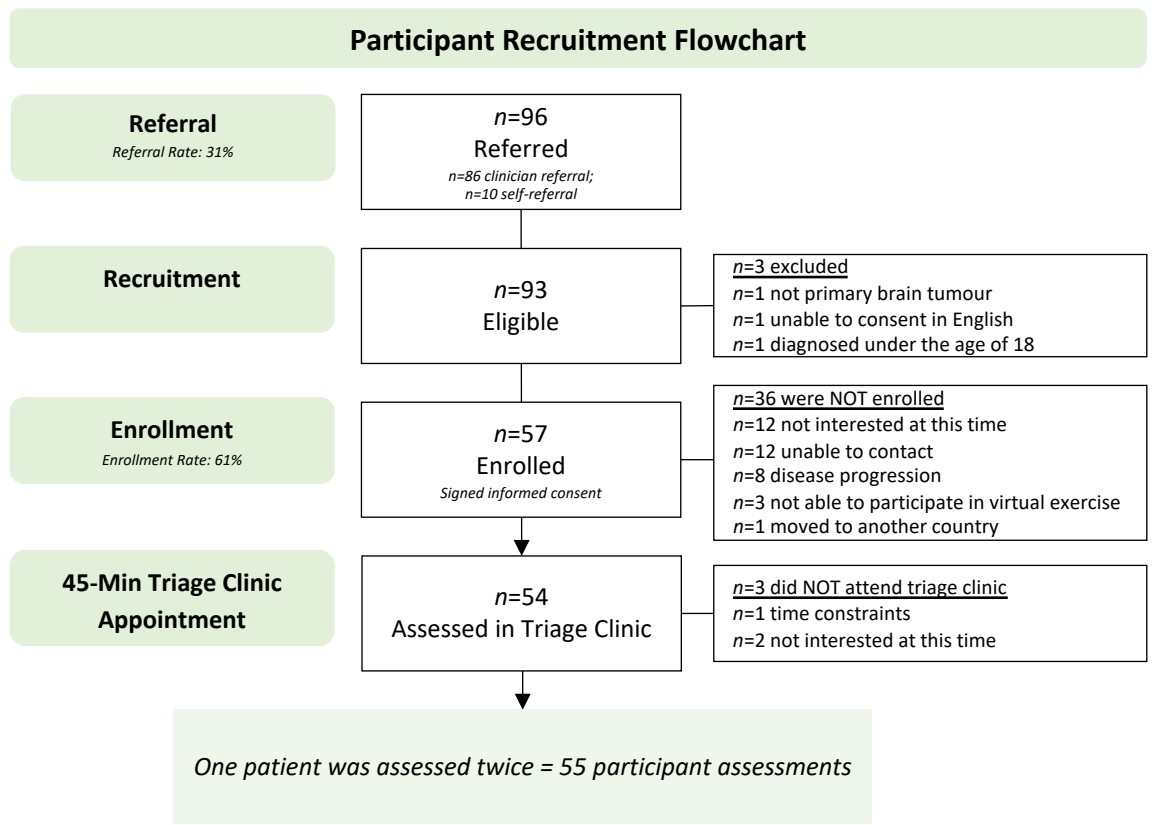

**Figure 3.** Participant recruitment flowchart.

*3.2. Triage Clinic Outcomes*

See Table 3 and Figure 4 for the triage clinic assessment results. Table 3 presents participants' vitals, body composition, and triage outcomes (i.e., SPPB, ECOG, and KPS scores). Appendix B includes the neurological examination results. Figure 4 reviews referral rates to the available rehabilitation and exercise resources. Of the 55 participant assessments, 49 met the inclusion criteria for exercise (i.e., SPPB $\geq$ 5, ECOG < 3, and KPS > 50) and were thus referred to the ACE-Neuro exercise intervention [25]. Six participants did not meet the initial criteria and were referred to either an individual ($n = 3$) or a combination ($n = 3$) of specialized rehabilitation services, including two referrals to physiatry, four to physiotherapy, and four to occupational therapy. Of the 49 referred to ACE-Neuro exercise, 22 of these were also referred to either one ($n = 19$) or multiple ($n = 3$) additional resources, including 5 referrals to physiatry, 5 to physiotherapy, and 15 to occupational therapy. The average BMI of triage clinic participants was 30.0 $\pm$ 6.5 kg/m$^2$. Resting heart rate and blood pressure were 80 $\pm$ 16 bpm and 122.8/83.2 mmHg, respectively. A total of 53 of the 55 participant assessments completed the full SPPB. Reasons for not completing the full or parts of the SPPB were related to safety (i.e., the triage clinic team or patient not feeling safe to complete) or an inability to perform (e.g., unable to ambulate). The mean SPPB score of patients was 8.9 $\pm$ 3.1. The majority of participants (57.1%) had an ECOG score of 1, with the next highest score being 2 (33.9%). A total of 91.1% of participants scored between 60 and 90 on the KPS, with 30.4% scoring 90, 17.9% scoring 80, 23.2% scoring 70, and 19.6% scoring 60. A total of 51 (92.7%) participants had deficits in the neurologic examination (i.e., four participants had completely normal exams). See Appendix B for full neurological examination results. Forty participants (72.7%) had cognitive deficits, 30 (54.5%) had deficits with cranial nerve examination, 11 (20.0%) had motor deficits, 25 (45.5%) had abnormal reflexes, 17 (30.9%) had peripheral sensory deficits, and 25 (45.5%) had coordination deficits. Eight participants had deficits only with cognition, but otherwise normal cranial nerve, motor, reflex, sensory, and coordination examinations.

**Table 3.** Triage clinic results—vitals, body composition, and triage criteria (SPPB, ECOG, and KPS), *n* = 55.

| Exam Component | Result (Mean $\pm$ SD) | |
|---|---|---|
| **Vitals** | | |
| Resting Heart Rate, bpm | 80 $\pm$ 16 | |
| Resting Blood Pressure, mm Hg | 122.8/83.2 | |
| Systolic Blood Pressure, mm Hg | 122.8 $\pm$ 16.2 | |
| Diastolic Blood Pressure, mm Hg | 83.2 $\pm$ 11.6 | |
| **Body Composition** | | |
| Height, kg | 169.2 $\pm$ 12.4 | |
| Weight, cm | 85.6 $\pm$ 20.4 | |
| BMI, kg/m$^2$ | 30.0 $\pm$ 6.5 | |
| **SPPB** | | |
| Balance Score, out of 4 | 2.9 $\pm$ 1.4 | |
| Gait Speed Score, out of 4 Gait Aids Used: walker (n = 4), cane (n = 5), none (n = 44) | 3.2 $\pm$ 1.1 | |
| Chair Stand Test Score, out of 4 | 2.7 $\pm$ 1.2 | |
| Total Score, out of 12 | 8.9 $\pm$ 3.1 | |
| **ECOG Score, 0–4 range** | **Number of Participants** | **Percentage, *n*/55 (%)** |
| 0 | 2 | 3.6 |
| 1 | 32 | 57.1 |
| 2 | 19 | 33.9 |
| 3 | 2 | 3.6 |
| 4 | 0 | 0 |
| **KPS Score, 100–0 range** | **Number of Participants** | **Percentage, *n*/55 (%)** |
| 100 | 1 | 0 |
| 90 | 17 | 30.4 |
| 80 | 10 | 17.9 |
| 70 | 13 | 23.2 |
| 60 | 11 | 19.6 |
| 50 | 3 | 5.4 |
| 40 | 1 | 1.8 |
| 30 | 0 | 0 |
| 20 | 0 | 0 |
| 10 | 0 | 0 |
| 0 | 0 | 0 |

SPPB: Short Physical Performance Battery; ECOG: Eastern Cooperative Oncology Group; and KPS: Karnofsky Performance Status. Triage criteria for exercise: SPPB < 5, ECOG > 3, and KPS < 50.

### 3.3. Qualitative Results

Of the 55 triage clinic participants, 20 completed a semi-structured interview. Of these 20 participants, four had caregivers present. In addition, five members of the clinical team completed an interview. Overall, all participants (i.e., participants and members of the clinical team) (1) felt satisfied with the triage clinic and (2) valued the triage clinic as part of neuro-oncology care. Appendix D includes additional representative quotes for these two categories.

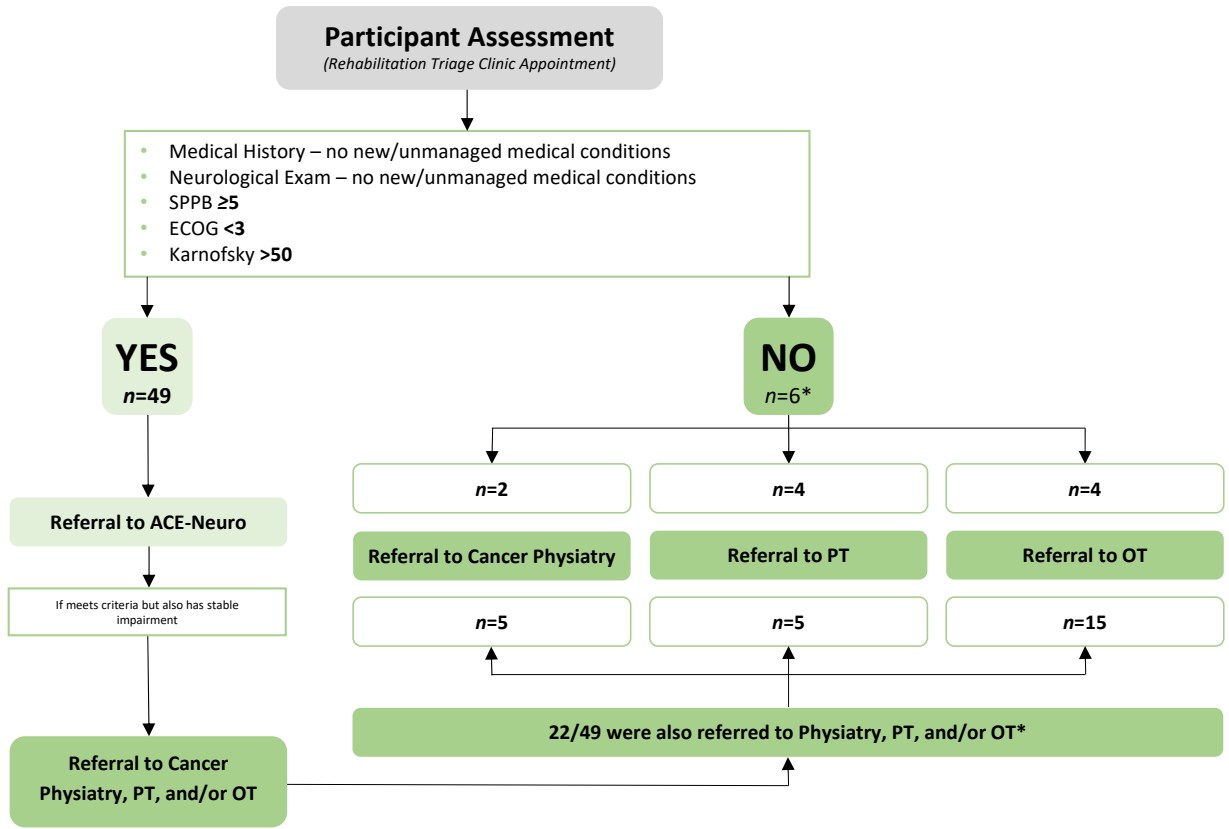

SPPB – Short Physical Performance Battery Protocol score; ECOG – Eastern Cooperative Oncology Group score; KPS – Karnofsky Performance Status score; PT – physiotherapy; OT – occupational therapy
*Some participants with referrals to multiple resources

**Figure 4.** Triage Clinic Results.

### 3.3.1. Category One: Satisfaction with the Rehabilitation Triage Clinic

Participants spoke of feeling satisfied with the triage clinic appointment safety, length, examination components (e.g., SPPB and neurological exam), personnel (i.e., resident physiatrist and exercise physiologist), and location. Participants also felt that attending the appointment in-person was feasible and helpful in advance of the subsequent ACE-Neuro exercise intervention (for those triaged to exercise).

> *This appointment was really very organized. I mean—when they informed me that I will be [ . . . ] that I need to do the assessment, it's very coordinated uh it's fast and then they're very warm and very supportive [ . . . ] I know that I'm in good hands because I know that they're gonna be supporting me. And [ . . . ] from the time that they contact you, the communication, the physical check-up, those are all, timed professionally and very organized. I love that they do that because it's more like knowing you a bit more based on what your situation [ . . . ] and seeing you before you do the activity is important so that they can assess your limitation as well.* Participant 04

Some participants spoke of feeling uncertain and nervous in advance of the triage clinic, but yet were ultimately satisfied with how the appointment was conducted.

> *Well, you know before you're kinda wondering what this is all about and you know you're more curious and once you get there, I think all of our questions were answered you were really good [at] taking us through that pre-assessment. I know there was a bit of a wait time there before you [ . . . ] decided whether you're in or out I thought, oh you know that might take longer I might have to go home and find out about it [ . . . ] in a week whatever, but you came right back and told us, so there was really no wait time and*

*we left with the equipment we needed, [ . . . ] so I uh I think it went yeah really quite smoothly.* Participant 07

Some participants shared feedback on ways to improve the triage clinic, including providing additional information on the rationale for the types of assessments chosen.

*It would have been nice to see [ . . . ] why you decided on those tests, and like the rationale so like we would know how it would be beneficial to us, because so far it kind of seemed like it was just a test to see if she was fit for the program.* Participant 37′s Caregiver

From the clinical team's perspective, referring to the triage clinic did not disrupt their clinical workflow and was thus perceived as a feasible addition to their neuro-oncology clinic.

*It is very easy just put in the order [for the referral to the triage clinic] the order is 2 seconds, so no, it seems like it's working.* Clinical Team Member 01 (Oncologist)

*I found it easy to refer. That was simple, even with [the new electronic medical record], it was easy to refer patients [ . . . ] I think patients, uhm, were seen a little bit faster than they were with just rehab, and I think their needs might have been more individualized and met.* Clinical Team Member 08 (Nurse)

Members of the clinical team also spoke to their satisfaction with the triage clinic personnel for triaging participants to an appropriate and tailored resource.

*I do like the triage system, because I know the patient would benefit from exercise and I know the patient would benefit from [occupational therapy] or [physiotherapy]. But it was nice having somebody who specializes in that area to make that decision.* Clinical Team Member 06 (Nurse Practitioner)

3.3.2. Category Two: Value of a Triage Clinic

Participants felt that the triage clinic was beneficial for providing them with a sense of hope in their cancer journey as well as for supporting access to additional resources.

*Well that there was maybe some hope [laughs] for getting some of these muscles working again [ . . . ] there's hope out there [ . . . ] it's not a dead-end.* Participant 43

*It was good it was great 'cause I finally got someone to—I finally got recognized. Well, not recognized, but you know, someone to actually help me out with [my brain cancer] so that's great.* Participant 17

*I thought that that was good, and out of that I ended up in occupational therapy as well as [ACE-Neuro], both of which were excellent programs and helped me.* Participant 51

*It opened up my eyes to some of the [resources and programs] that were available to me that I didn't even know about.* Participant 59

*It was probably the best day I've had in a really long time. Having [the triage clinic], be truly kindness, and an opening to just whatever I needed. You guys were there, period. You were there, and you never talked to each other like I wasn't part of it. So, everything that was brought up was brought up for all of us to be part of which I thought was kindness, and just an openness that made it UN scary, which was lovely [ . . . ] For me that was one of the best [appointments] that I've been- Not one of, that was the best I've been to of an appointment. Yeah, that was I above and beyond . . . that was perfect for me.* Participant 52

Members of the clinical team felt that referring to the triage clinic was beneficial for participants for supporting safety in advance of exercise participation (for those triaged to exercise), as well as for patient experience by needing only one referral per patient. Further, members of the clinical team felt that referring to just one source also simplified their referral process and workflow in the clinic.

*I think that simplifies things for us a lot right? So one, it is a one-point of referral. And then you guys do the bulk work, really? And sometimes we refer, and I've heard that*

*we refer to physiatry, but then the team feels that the patient should be really seen by [occupational therapy]. [ … ] Sometimes we are not sure who to refer the patient to, and what would be the best fit, so I think that was quite nice to be just able to, you know, refer to rehab, and then see what's the best for the patient.* Clinical Team Member 07 (Oncologist)

*You need to do the triage, I think, That's what [makes] it safe [ … ] you need that triage to know what the patient is appropriate for.* Clinical Team Member 01 (Oncologist)

Finally, members of the clinical team spoke about the possibilities of a triage clinic that extends beyond the neuro-oncology patient population.

*I would like to see it grow beyond brain tumours, I know [the research team] is looking at head and neck as well but is there a role and vision for a triage clinic to assess rehab readiness for everyone with a cancer diagnosis? There could be many more layers to this clinic..* Clinical Team 4.

## 4. Discussion

The concept of cancer rehabilitation and exercise was first introduced over 40 years ago, with barriers at that time including difficulty identifying patients in need, and awareness from oncologists on the role of rehabilitation and activity [40]. Unfortunately, these same barriers exist today [12,15,16]. With improving survival rates among cancer patients, the role of functional rehabilitation and exercise is more important than ever [1]. Cancer survivors report long-term concerns with function, quality of life, and inactivity following their diagnosis [2,3]. To date, consistent screening for inactivity and impairment, as well as triage and referral pathways (i.e., through the EMR) to appropriate rehabilitation and exercise resources (i.e., physiatry, physiotherapy, occupational therapy, and exercise), do not exist in most cancer care systems.

Over the last several years, multiple researchers and clinicians have identified the critical need for improved impairment-driven cancer rehabilitation [15,16]. Screening for distress programs, including the revised Edmonton System Assessment System (ESAS) and the Canadian Problem Checklist, have been implemented in most Canadian Cancer Centers [19,20]. The purpose of these pre-existing tools is to help healthcare providers identify, assess, and manage distressing symptoms and concerns experienced by patients, and enhance the person centeredness of care delivered by providing appropriate and tailored referrals [31]. The purpose is also to have automated thresholds that trigger referrals to appropriate resources, avoiding missed opportunities for patient care. These tools screen for symptoms like nausea, fatigue, and shortness of breath, but do not include critical screening questions related to activities of daily living, physical function, or activity levels. The Screening for Distress initiative was based on research showing the profound benefit of routine screening for distress among patients and the value of referring to appropriate resources within the cancer care setting as needed [20]. Recent studies indicate that more cancer survivors report decreased health-related quality of life related to physical impairment versus psychological impairment, begging the need for improved research and implementation of screening, triage, and referral for physical impairment in addition to psychological impairment [41,42]. Early research to develop rehabilitation care pathways are underway in the United States, with more work necessary to develop and test screening and clinical referral pathways that will better serve cancer patients worldwide [14,15].

Neuro-oncology patients have unique needs, with impairments often affecting function, including cognition, mobility, and coordination [26]. The purpose of this study was thus to assess the feasibility of a triage clinic to define common impairments or deficits among neuro-oncology patients and assess the feasibility of triage decision making and referral to both rehabilitation and exercise resources. Overall, we found that the triage clinic was feasible from an enrollment and attendance perspective, based on achieving pre-determined cut-offs and based on participant qualitative reports on the enrollment pathway. To contribute to overall feasibility, we importantly found that the triage clinic was

safe, with no adverse events during the triage clinic appointment. Participants commented that the assessments were organized and thorough. Finally, the triage clinic was found to be feasible based on the appropriate triage of participants to rehabilitation and exercise services using the pre-determined triage tools.

The enrollment rate of 61% exceeded our a priori feasibility rate of 50%, and the triage clinic attendance rate of 94.7% also exceeded our a priori feasibility, set at 60%. On average, individuals were seen in the triage clinic 22.2 business days after their referral, which from the qualitative data, was deemed acceptable by both participants and clinicians. Further, participants spoke about the value of the triage clinic as a part of their neuro-oncology care, commenting on the in-depth assessment that informed their access to appropriate rehabilitation resources in a timely manner. Participants felt the clinic offered a tailored approach to their rehabilitation care. Clinical team members commented on how the triage clinic simplified their referral processes, feeling that they could refer to one place and their patients would be further assessed to determine specific rehabilitation needs. One clinical team member commented on how they would like the triage clinic to grow beyond brain tumours and into other tumour groups. Overall, these quantitative and qualitative results support the feasibility of enrollment and triage clinic attendance for the neuro-oncology population, as well as the acceptability of the triage clinic appointment.

The pre-determined tools used for the triage decision included a health history screening interview, a neurological examination, the SPPB, ECOG, and KPS. Importantly, 93% of participants assessed in the clinic had a neurological deficit (i.e., 51 out of 55 participants). The most common deficits were with cognition, cranial nerves, reflexes, and coordination. These triage clinic results clearly show the prevalence of neurological deficits often contributing to patient functional impairment, and point to the need for triage to resources that are appropriate and tailored to each patient's needs. Appropriate triage can support streamlined access to rehabilitation and exercise resources in a timely fashion, without participants having to be re-referred to separate providers across multiple visits.

Functionally, participants, on average, scored $8.9 \pm 3.1$ on the SPPB out of 12. The previous literature on frailty suggests a score of lower than 10 indicates one or more mobility limitations and is predictive for all-cause mortality [43,44]. Therefore, the pre-determined cut-off to be eligible for the ACE-Neuro exercise study was initially 10/12; however, this was changed to 5/12 after the first five participants were assessed. It was clear that due to balance issues, gait speed, and decreased leg endurance, the majority of scores were less than 10/12. Despite this scoring and one or more mobility limitations, participants were still able to perform basic chair exercises, making them eligible for the ACE-Neuro exercise study. For this reason, the criteria were changed to ensure participants who were frail or had more than one mobility limitation were not excluded from the ACE-Neuro exercise study. Those scoring below 5/12 often required mobility aids and therefore did not meet the eligibility criteria for the ACE-Neuro exercise study. Of those who did not meet eligibility criteria on the SPPB for the ACE-Neuro exercise study ($n = 5$), the barriers were mainly not being able to complete one or more of the three tests (i.e., balance, gait speed, and chair to stand). From a clinical feasibility perspective, the SPPB was an easy assessment to administer and was tolerated well by participants.

The KPS and ECOG scores were determined by the physiatry resident and clinical exercise physiologist based on health history, neurological examination, and the SPPB. The majority of patients scored 1 on the ECOG (57.1%), i.e., "restricted in physically strenuous activity but ambulatory and able to carry out work for a light or sedentary nature" [35]. On the KPS, the majority of scores were distributed between 60 and 90/100, with the largest group scoring 90 (30.4%, i.e., "able to carry on normal activity with minor signs or symptoms of disease") and the next largest group scoring 70 (23.3%, i.e., "cares for self but unable to carry on normal activity or do active work") [34]. Moving forward, selecting one of these functional status scores would be reasonable as they provide similar data. The KPS, which has more data intervals compared to the ECOG, allows for a more specific categorization of function, which may help to facilitate referral decisions more easily. Using

the triage clinic criteria, a total of 49 participants were referred to the ACE-Neuro exercise study, and of these, 22 participants required additional rehabilitation services referrals to address specific impairments (See Figure 4). Overall, participants found attending the triage clinic feasible and beneficial.

Interestingly, the referral rate into the study was 31%, which was less than the a priori feasibility level of 50%. One reason patients were not referred was due to "clinical judgement" by the oncologist or nurse in the neuro-oncology clinic. Potential barriers to referral amongst the 69% not referred may have included the perception that rehabilitation and exercise were not necessary or not medically appropriate for the majority of patients. However, previous research in other tumour groups has shown that physical impairment impacts over 90% of patients [17], and our results show motor or sensory impairment amongst 92.7% of participants assessed. Cheville and colleagues found that while 91% of patients reported needing rehabilitation services post-diagnosis, only 30% reported receiving this care [17]. Other reports suggest physical rehabilitation needs rank highest in unmet needs, over financial, emotional, communication, body image, and multiple other categories of needs, and that physical impairment is a key contributor to psychosocial distress [41,42]. In addition, a lack of screening and identification is a significant cause of high physical impairment rates among patients [16]. To address this in the future, improved patient screening and ease of referral to rehabilitation resources (i.e., through an EMR), as well as education for healthcare providers, may be a means to increase referral rates within standard clinical care.

Overall, this study highlights the lack of standardized identifications of patients with functional impairment or who are currently sedentary. Once patients are identified, however, our triage clinic results indicate that effective and efficient assessment, triage, and referral of these patients to appropriate rehabilitation resources is feasible and well accepted both by patients and clinical team members. To improve the identification of functional impairment among patients, we thus propose a tool for screening called the Cancer Rehabilitation and Exercise Screening Tool (CREST, see Figure 5). This simple assessment takes less than 5 min to complete and can assist with identifying the most common functional impairments seen in individuals living with and beyond cancer. CREST was developed by cancer physiatrists, cancer and exercise researchers, physicians, and exercise physiologists, and can be implemented within the Cancer Exercise and Rehabilitation Pathways Model (see Figure 6), adapted from our prior work with colleagues [21,22]. The proposed CREST tool screens for physical inactivity and allows participants to report pre-identified functional concerns and difficulties with activities of daily living using a 1–10 Likert scale. Similar to the ESAS, which has now been widely implemented at most cancer appointments [31], CREST may improve the efficient and effective identification of those with functional impairment. To the best of our knowledge, no other functional screening tool designed for implementation in a clinical setting has been successfully integrated into cancer care. This is despite reports that a screening tool would help to better identify patients with impairment, potentially improving patient care and recovery [16]. Research tools like the Functional Assessment of Cancer Therapy scale (FACT) and the SF-36 exist, but are not designed for screening purposes (i.e., the FACT and SF-36) and/or are not specific to cancer (i.e., SF-36) [45,46]. The Functional Independence Measure (FIM) is a well-validated measure for disability, but it is not designed as a screening tool and is not validated in the cancer population [47]. Recently the Patient-Report Outcomes Measurement Information System (PROMIS) Cancer Function Brief 3D profile has been proposed as a composite of three short forms that evaluate gross and upper extremity function, fatigue, social participation, cognition, and fine motor skills, but it is not designed to identify specific impairments that can aid in triage and referral to specific rehabilitation specialists [48,49]. In addition, it was originally designed as a research tool, although more recent reports have investigated its role as a clinical tool [48,49]. The CREST, specifically designed as an in-clinic screening tool, may be used at each oncology appointment to identify new or existing functional impairments among patients. The tool can be completed in the waiting room by patients and reviewed with the clinical team members or healthcare providers, who can then facilitate

appropriate referrals to either a triage clinic for further assessment, or directly to specific resources (i.e., physiatry, physiotherapy, or occupational therapy) for those with functional impairments. For those without any current impairment but who are inactive, a referral to exercise resources can be made. For individuals meeting activity guidelines without any impairment, they may only need to receive electronic or printed resources to support the maintenance of their active lifestyles. The hope is that with improved screening, we can close the gap between those with functional rehabilitation or inactivity concerns and those referred to rehabilitation and exercise resources. Future studies are necessary to validate and assess the benefit and implementation of the CREST.

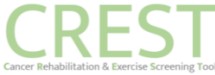

## CREST: Cancer Rehabilitation & Exercise Screening Tool

**Patient information:**
- **Patient Name**: _________________________________
- **Date Completed**: _________________________ **Time Completed**: _____________________
- **Completed By** (check one): ☐ Patient ☐ Caregiver ☐ Caregiver-Assisted ☐ Healthcare Provider

**On average, in the last week:**

I participated in moderate (i.e., brisk walking) *OR* vigorous (i.e., running) **cardiovascular exercise** ______ (a) times per week, for on average ______(b) minutes = ______(a x b) **TOTAL WEEKLY AEROBIC MINUTES**
**(≥90 MINUTES = MEETING GUIDELINES)**

I participated in ______(c) sessions of **resistance exercise** = ______(c) **TOTAL RESISTANCE EXERCISE SESSIONS**
**(≥2 SESSIONS PER WEEK = MEETING GUIDELINES)**

**MEETING EXERCISE GUIDELINES? _____ YES _____ NO**

**Please circle the number that best describes your symptoms in the last 7 days:**

| | NO CONCERN | | | | | | | | | MAJOR CONCERN |
|---|---|---|---|---|---|---|---|---|---|---|
| Vision changes | 1 | 2 | 3 | 4 | 5 | 6 | 7 | 8 | 9 | 10 |
| Balance issues | 1 | 2 | 3 | 4 | 5 | 6 | 7 | 8 | 9 | 10 |
| Swallowing issues | 1 | 2 | 3 | 4 | 5 | 6 | 7 | 8 | 9 | 10 |
| Speech issues | 1 | 2 | 3 | 4 | 5 | 6 | 7 | 8 | 9 | 10 |
| Numbness/tingling | 1 | 2 | 3 | 4 | 5 | 6 | 7 | 8 | 9 | 10 |
| Joint pain | 1 | 2 | 3 | 4 | 5 | 6 | 7 | 8 | 9 | 10 |
| Back pain | 1 | 2 | 3 | 4 | 5 | 6 | 7 | 8 | 9 | 10 |
| Arm/ leg pain | 1 | 2 | 3 | 4 | 5 | 6 | 7 | 8 | 9 | 10 |
| Fatigue | 1 | 2 | 3 | 4 | 5 | 6 | 7 | 8 | 9 | 10 |
| Falls | 1 | 2 | 3 | 4 | 5 | 6 | 7 | 8 | 9 | 10 |
| Difficulty getting up and down stairs | 1 | 2 | 3 | 4 | 5 | 6 | 7 | 8 | 9 | 10 |
| Difficulty getting up from a chair | 1 | 2 | 3 | 4 | 5 | 6 | 7 | 8 | 9 | 10 |
| Memory or cognitive changes | 1 | 2 | 3 | 4 | 5 | 6 | 7 | 8 | 9 | 10 |

**In the last 7 days, how difficult has it been for you to complete the following activities independently:**

| | EASY | | | | | | | | | VERY DIFFICULT/NEED ASSISTANCE |
|---|---|---|---|---|---|---|---|---|---|---|
| Dressing | 1 | 2 | 3 | 4 | 5 | 6 | 7 | 8 | 9 | 10 |
| Eating | 1 | 2 | 3 | 4 | 5 | 6 | 7 | 8 | 9 | 10 |
| Walking | 1 | 2 | 3 | 4 | 5 | 6 | 7 | 8 | 9 | 10 |
| Toileting | 1 | 2 | 3 | 4 | 5 | 6 | 7 | 8 | 9 | 10 |
| Bathing | 1 | 2 | 3 | 4 | 5 | 6 | 7 | 8 | 9 | 10 |
| Walking 1 block | 1 | 2 | 3 | 4 | 5 | 6 | 7 | 8 | 9 | 10 |

**Figure 5.** Cancer Rehabilitation and Exercise Screening Tool (CREST).

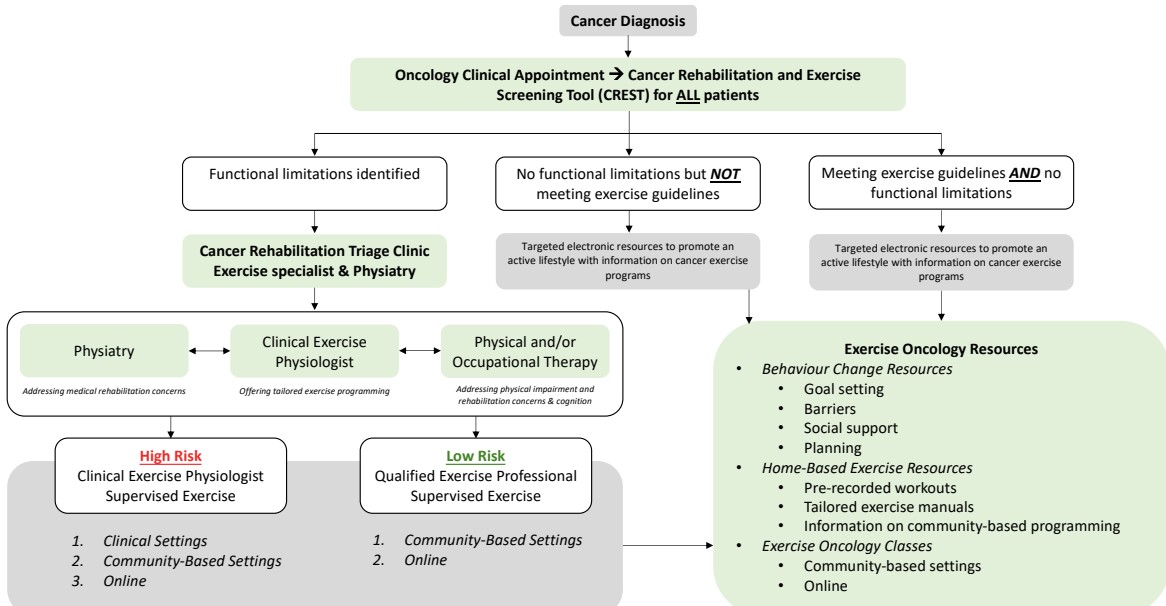

**Figure 6.** Cancer Rehabilitation and Exercise Pathways Model. Adapted from Santa Mina et al., 2018, [21] and Wagoner et al., 2022 [22].

As Smith and colleagues stated, "it is challenging, if not impossible, to imagine a high-quality oncology care system that does not include rehabilitation services" [6]. Evidence supports the role of cancer rehabilitation, which includes screening for functional impairment and inactivity, as a way to improve function and quality of life among patients [16]. Therefore, work is needed to both improve the identification of patients with functional impairment, and the triage and referral of these patients to appropriate services. The triage clinic results indicate that the recruitment of patients is difficult, likely due to a lack of consistent screening and identification of those in need [6]. Our hope is that CREST will be implemented within the Cancer Rehabilitation and Exercise Pathways Model as a screening resource, and the triage clinic will provide assessment for complex patients, allowing for referral to the right rehabilitation and exercise resources at the right time. With improved screening, triage and referral into rehabilitation resources, those living with and beyond cancer have the potential to more easily access the support they need, improving their recovery and quality of life into survivorship.

**Author Contributions:** Conceptualization, L.C.C., J.T.D., G.J.F., M.d.G.W., G.R.U. and N.C.-R.; methodology, L.C.C., J.T.D., G.J.F., M.d.G.W. and N.C.-R.; formal analysis, L.C.C., J.T.D. and N.C.-R.; writing—original draft preparation, L.C.C., J.T.D. and N.C.-R.; writing—review and editing, L.C.C., J.T.D., G.J.F., M.d.G.W., G.R.U., N.C.-R. and D.L.; funding acquisition, N.C.-R. All authors have read and agreed to the published version of the manuscript.

**Funding:** This project was supported by the Alberta Cancer Foundation (Grant number: 26047) under the Principal Investigator, Nicole Culos-Reed.

**Institutional Review Board Statement:** The study was conducted in accordance with the Declaration of Helsinki and approved by the Calgary Health Research Ethics Board of Alberta (HREBA)—Cancer Committee (CC)—HREBA.CC-20-0322.

**Informed Consent Statement:** Informed consent was obtained from all subjects involved in the study.

**Data Availability Statement:** Data are stored at the University of Calgary.

**Acknowledgments:** We would like to acknowledge Emma McLaughlin, Mannat Bansal, and Tana Dhruva for their support in data reporting for this manuscript.

**Conflicts of Interest:** The authors declare no conflict of interest. The funders had no role in the design of the study; in the collection, analyses, or interpretation of data; in the writing of the manuscript, or in the decision to publish the results.

## Appendix A

**Table A1.** Participant co-morbidities and cancer-related side-effects.

| Co-Morbidities | No. of Patients |
| --- | --- |
| Anxiety/Depression | 14 |
| Arthritis | 4 |
| Asthma | 3 |
| Auditory Impairments | 6 |
| Benign Prostatic Hyperplasia | 1 |
| Blood Disorders | 13 |
| Cardiovascular Disease | 6 |
| Chronic Sinusitis | 1 |
| Concussion | 2 |
| Diabetes | 3 |
| Dyslipidemia | 7 |
| Gastrointestinal Disorders | 14 |
| Headaches/Migraine | 2 |
| Hypercalcemia | 1 |
| Hypertension | 10 |
| Hypotension | 1 |
| Infectious Disease | 1 |
| Kidney Disease | 1 |
| Major Laceration | 1 |
| Musculoskeletal Pain/Injuries | 44 |
| Neurodevelopmental Disorder | 1 |
| Neurological Disease: Multiple Sclerosis, Restless Leg Syndrome | 2 |
| Obstructive Sleep Apnea | 6 |
| Optic Issues | 2 |
| Osteoarthritis | 2 |
| Osteoporosis | 1 |
| Other Cancers | 8 |
| Peripheral Vascular Disease | 1 |
| Previous Surgeries | 66 |
| Pulmonary Conditions | 1 |
| Skin Conditions | 3 |
| Stroke | 3 |
| Thyroid Disease | 7 |
| Viral Disease | 4 |
| **Cancer-Related Side-Effects** | **No. of Patients** |
| Ataxia | 2 |
| Balance Challenges | 31 |
| Cognition Challenges | 35 |
| Constipation | 2 |
| Decreased Mobility | 1 |
| Dizziness | 10 |
| Dysphagia | 2 |
| Dyspnea | 8 |
| Fainting | 2 |
| Fatigue | 25 |
| Headaches | 22 |
| Hearing Challenges | 3 |

**Table A1.** *Cont.*

| Cancer-Related Side-Effects | No. of Patients |
|---|---|
| Hemiplegia | 1 |
| Incontinence | 1 |
| Incoordination | 3 |
| Increased Wound Healing Time | 1 |
| Itchiness | 1 |
| Loss of Appetite | 1 |
| Major Neurological Disorder | 2 |
| Mood Changes | 1 |
| Nausea or Vomiting | 1 |
| Neglect | 2 |
| Numbness/Tingling | 13 |
| Obstructive Hydrocephalus | 1 |
| Osteopenia | 1 |
| Pain | 2 |
| Seizures | 35 |
| Sensory Deficits | 2 |
| Spasticity | 1 |
| Speech Challenges | 6 |
| Tinnitus | 7 |
| Tremors | 1 |
| Vision Challenges | 16 |
| Voice Changes | 1 |
| Weakness | 32 |
| Writing Challenges | 1 |

## Appendix B

**Table A2.** Triage clinic results—neurological examination, $n = 55$.

| Exam Component | No. With Deficiency $n$/55 (%) | No. Unable to Perform $n$/55 (%) |
|---|---|---|
| Orientation | | |
| Date | 16 (29.1) | 1 (1.8) |
| Location | 6 (10.9) | 1 (1.8) |
| Attention, Registration, and Recall | | |
| Serial 7s | 21 (38.2) | 4 (7.3) |
| "World" spelled backward | 14 (25.5) | 3 (5.5) |
| Recall 3 objects (red, truck, and velvet) | 14 (25.5) | 4 (7.3) |
| Language | | |
| Name three objects (pen, watch, and glasses) | 5 (9.1) | 1 (1.8) |
| Repeat: "no ifs, ands, or buts" | 4 (7.3) | 1 (1.8) |
| Complete a three-stage command | 0 (0) | 2 (3.6) |
| Read a sentence | 1 (1.8) | 1 (1.8) |
| Write a sentence | 3 (5.5) | 1 (1.8) |
| Draw a pentagon | 4 (7.3) | 1 (1.8) |

**Table A2.** *Cont.*

| Exam Component | No. With Deficiency *n*/55 (%) | No. Unable to Perform *n*/55 (%) |
|---|---|---|
| Cranial Nerve Exam | | |
| Cranial Nerve II | | |
| Pupils equal and reactive to light and accommodating | 1 (1.8) | 0 (0) |
| Fields intact | 5 (9.1) | 0 (0) |
| Cranial Nerve III, IV, and VI | | |
| Extraocular movements intact | 1 (1.8) | 0 (0) |
| Ptosis | 4 (7.3) | 0 (0) |
| Nystagmus | 6 (10.9) | 0 (0) |
| Diplopia | 2 (3.6) | 0 (0) |
| Pursuit | 4 (7.3) | 0 (0) |
| Saccades | 3 (5.5) | 0 (0) |
| Cranial Nerve V | | |
| Sensation intact—V1 | 4 (7.3) | 0 (0) |
| Sensation intact—V2 | 3 (5.5) | 0 (0) |
| Sensation Intact—V3 | 4 (7.3) | 0 (0) |
| Masseters, Pterygoids, and Temporalis | 0 (0) | 0 (0) |
| Cranial Nerve VII | | |
| Wrinkle forehead | 3 (5.5) | 0 (0) |
| Eye closure | 2 (3.6) | 0 (0) |
| Smile | 0 (0) | 0 (0) |
| Cranial Nerve VIII | | |
| Hearing | 11 (20) | 0 (0) |
| Cranial Nerve IX and X | | |
| Dysarthria | 0 (0) | 0 (0) |
| Uvula midline | 1 (1.8) | 0 (0) |
| Soft palate rise | 1 (1.8) | 0 (0) |
| Cranial Nerve XI | | |
| Sternocleidomastoid strength | 1 (1.8) | 0 (0) |
| Trapezius strength | 4 (7.3) | 1 (1.8) |
| Cranial Nerve XII | | |
| Hypoglossal | 0 (0) | 0 (0) |
| Motor | | |
| Bulk intact, upper body | 3 (5.5) | 0 (0) |
| Bulk intact, lower body | 1 (1.8) | 0 (0) |
| Tone | 7 (12.7) | 0 (0) |
| Normal, n = 48 | | |
| Spasticity, n = 6 | | |
| Hypotonia, n = 1 | | |
| Power | | |
| Deltoids—(L) | 9 (16.4) | 0 (0) |
| Deltoids—(R) | 1 (1.8) | 1 (1.8) |
| Biceps—(L) | 6 (10.9) | 0 (0) |
| Biceps—(R) | 1 (1.8) | 1 (1.8) |
| Triceps—(L) | 8 (14.5) | 0 (0) |
| Triceps—(R) | 1 (1.8) | 1 (1.8) |
| Wrist extension—(L) | 9 (16.4) | 0 (0) |
| Wrist extension—(R) | 1 (1.8) | 0 (0) |
| Finger extension—(L) | 9 (16.4) | 0 (0) |

**Table A2.** *Cont.*

| Exam Component | No. With Deficiency *n*/55 (%) | No. Unable to Perform *n*/55 (%) |
|---|---|---|
| Finger extension—(R) | 1 (1.8) | 0 (0) |
| Finger flexion—(L) | 7 (12.7) | 0 (0) |
| Finger flexion—(R) | 0 (0) | 0 (0) |
| Hand intrinsics—(L) | 7 (12.7) | 0 (0) |
| Hand intrinsics—(R) | 2 (3.6) | 0 (0) |
| Hip flexion—(L) | 11 (20) | 0 (0) |
| Hip flexion—(R) | 6 (10.9) | 0 (0) |
| Hip extension—(L) | 5 (9.1) | 1 (1.8) |
| Hip extension—(R) | 1 (1.8) | 1 (1.8) |
| Knee flexion—(L) | 7 (12.7) | 0 (0) |
| Knee flexion—(R) | 2 (3.6) | 0 (0) |
| Knee extension—(L) | 9 (16.4) | 0 (0) |
| Knee extension—(R) | 3 (5.5) | 0 (0) |
| Ankle dorsiflexion—(L) | 9 (16.4) | 0 (0) |
| Ankle dorsiflexion—(R) | 1 (1.8) | 0 (0) |
| Ankle plantarflexion—(L) | 6 (10.9) | 0 (0) |
| Ankle plantarflexion—(R) | 0 (0) | 0 (0) |
| Toe extension—(L) | 6 (10.9) | 0 (0) |
| Toe extension—(R) | 2 (3.6) | 0 (0) |
| Reflexes | | |
| Biceps C5, C6—(R) | 5 (9.1) | 0 (0) |
| Biceps C5, C6—(L) | 13 (23.6) | 0 (0) |
| Triceps C6, C7, C8—(R) | 2 (3.6) | 0 (0) |
| Triceps C6, C7, C8—(L) | 13 (23.6) | 0 (0) |
| Brachioradialis C5, C6—(R) | 3 (5.5) | 0 (0) |
| Brachioradialis C5, C6—(L) | 14 (25.5) | 0 (0) |
| Hoffman T1—(R) | 4 (7.3) | 0 (0) |
| Hoffman T1—(L) | 3 (5.5) | 0 (0) |
| Knee L2/3/4—(R) | 3 (5.5) | 0 (0) |
| Knee L2/3/4—(L) | 9 (16.4) | 0 (0) |
| Ankle S1, S2—(R) | 2 (3.6) | 0 (0) |
| Ankle S1, S2—(L) | 6 (10.9) | 1 (1.8) |
| Plantar L4/5, S1/2—(R) | 1 (1.8) | 0 (0) |
| Plantar L4/5, S1/2—(L) | 1 (1.8) | 1 (1.8) |
| Clonus—(R) | 1 (1.8) | 0 (0) |
| Clonus—(L) | 2 (3.6) | 0 (0) |
| Sensory | | |
| Upper Extremity | | |
| Pinprick | 8 (14.5) | 0 (0) |
| Light touch | 16 (29.1) | 0 (0) |
| Lower Extremity | | |
| Pinprick | 5 (9.1) | 0 (0) |
| Light touch | 12 (21.8) | 0 (0) |
| Coordination | | |
| Finger to nose—(R) | 7 (12.7) | 1 (1.8) |
| Finger to nose—(L) | 13 (23.6) | 2 (3.6) |
| Heel to shin—(R) | 4 (7.3) | 1 (1.8) |
| Heel to shin—(L) | 11 (20) | 1 (1.8) |
| Rapid alternating movements—(R) upper extremity | 8 (14.5) | 2 (3.6) |
| Rapid alternating movements—(L) upper extremity | 10 (18.2) | 3 (5.5) |
| Rapid alternating movements—(R) lower extremity | 6 (10.1) | 1 (1.8) |
| Rapid alternating movements—(L) lower extremity | 11 (20) | 1 (1.8) |
| Fine motor coordination—(R) | 6 (10.1) | 1 (1.8) |
| Fine motor coordination—(L) | 11 (20) | 1 (1.8) |

(L): left side; (R): right side; and *n* = 54 participants, *n* = 1 re-referral seen in clinic twice.

## Appendix C

**Table A3.** Examples of interview questions.

| | |
|---|---|
| Examples of Interview Questions for Participants (i.e., neuro-oncology patients) | • *How was your experience with the triage clinic appointment?*<br>• *How did you find the duration of the appointment?*<br>• *How did you find the assessments performed during the appointment?*<br>• *In what ways did the triage clinic support your access to rehabilitation and exercise resources?*<br>• *How did you find the role of the physiatry resident and exercise physiologist?*<br>• *How did you find the safety of the appointment?*<br>• *How could this triage clinic be improved?*<br>• *How do you feel about the timing of this assessment from your diagnosis date?* |
| Examples of Interview Questions for Members of the Clinical Team | *For the ACE-Neuro study, recruitment took place* via *direct referral from you/the clinical team to the research team* via *the Putting Patients First Questionnaire in the Electronic Medical Record. Would you please tell me about your experience with this process?*<br>• *What worked well with this process?*<br>• *What changes did you notice with patient flow and access to resources with this process?*<br>• *What challenges did you experience with this process?*<br>• *How did this process impact your clinical workflow?*<br>• *What are your thoughts on the role of a triage clinic for neuro-oncology patients?*<br>• *In what capacity do you see this process working long-term/as part of standard care?* |

## Appendix D

**Table A4.** Additional participant quotes from qualitative analysis.

| Category | Participant Quotes | Clinical Team Member Quotes |
|---|---|---|
| (1) Satisfaction with the rehabilitation triage clinic | *[The triage clinic components] were not overly intense. Just like fine for me again in my current physical shape. Yeah. And I think even way back in treatment days, I probably wouldn't have had much trouble with what I was being asked of. Participant 65* | *It was just 'click the button' and it was right there, ACE-Neuro [laughs], it was easy [ . . . ] it was really easy. Clinical Team Member 08 (Nurse)* |
| (2) Value of a rehabilitation triage clinic | *I already had my occupational therapist that I was seeing and so it was like having you guys supply these things was great because then I didn't have to search for it. I didn't have to find one. I didn't have to tell my Mom whether or not I was open for it. Yeah, I think it was great for you guys to have that many options. And that many things happening. Participant 52* | *My wait time for consult 8 weeks right? So, I think that the triage clinic needs to play a bit of a role where it's way finding so patients get timely access to care. Clinical Team Member 02 (Rehabilitation Manager)*<br>*I find that the triage can be really helpful, or it can just create extra work unnecessarily. Like you know everyone benefits from exercise and you don't want to delay someone from starting exercise you almost want to empower them. Say you're ready to go, but there is a group of people that you need to kind of coach and assess to really clear them, and that's probably where this this triage clinic fits, right? Clinical Team Member 02 (Rehabilitation Manager)*<br>*After that it's I like the idea of the triage system because then I wasn't making you know the decision on what exactly they need. I mean, you know I refer to physio, OT, and to the exercise program all the time, but putting it through the triage, I knew they would get an assessment, and then it would be determined by somebody who specializes in rehab what would benefit the patient the most without me just saying I think you need all of this or you need this or . . . .So I like that process. And actually, you know, having access to PT, OT, physiatry, kind of all-in-one spot. I don't know if there was—I like the idea of the triage to actually, you know, for rehab to make a decision on a program that would be more beneficial for the patient. Clinical Team Member 06 (Nurse Practitioner)* |

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
