# Peer review of "Feasibility and Implementation of an Oncology Rehabilitation Triage Clinic: Assessing Rehabilitation, Exercise Need, and Triage Pathways within the Alberta Cancer Exercise–Neuro-Oncology Study"

_curroncol, doi:10.3390/curroncol30070461_

Round 1

Reviewer 1 Report

The abstract concisely summarizes the study, outlining the purpose, methodology, and critical findings. It offers insights into the feasibility and effectiveness of the TC in facilitating rehabilitation and exercise referrals for patients with brain tumors. The mention of the proposed screening tool indicates a potential strategy for improving the identification of patients needing rehabilitation and exercise interventions.

Few areas that could be improved:

•           Background and context: The abstract could provide more background information to contextualize the study. It would be beneficial to briefly discuss the existing literature on physical impairment and sedentary behavior in neuro-oncology populations.

•           Referral rates explanation: The abstract mentions that referral rates into the Triage Clinic (TC) were not met, but it does not explain this outcome clearly. Providing some insights into the potential reasons for the lower-than-expected referral rates would enhance the understanding of the study's limitations

•           Impact and implications: The abstract could expand on the implications and potential impact of the study. It would be helpful to discuss how the findings of the feasibility assessment and the proposed screening tool (CREST) could influence clinical practice and improve patient outcomes.

Introduction:

•           The introduction provides sufficient background information and includes relevant references.

•           The introduction can be improved by reorganizing the content for better flow and coherence. In addition, the information is currently presented in a somewhat fragmented manner.

•           Some sentences could be rephrased for clarity and readability.

•           It would be helpful to provide a clear statement of the research objectives or research question early in the introduction to guide the reader.

For example:

•           Reorganizing the content for better flow and coherence: Original sentence: "These patients often report ongoing cognitive, physical, and psychological impairment that is not adequately addressed, and are currently an understudied population in rehabilitation oncology literature [5, 6]." Revised sentence: "Neuro-oncology patients, who frequently experience cognitive, physical, and psychological impairment, often find that their needs are not adequately addressed. Surprisingly, this patient population remains understudied in rehabilitation oncology [5, 6]."

•           Rephrasing for clarity and readability: Original sentence: "Once patients are screened and identified as having functional deficits or concerns, triage and referral pathways are necessary." Revised sentence: "Following the identification of patients with functional deficits or concerns through screening, it becomes essential to establish triage and referral pathways."

•           Providing a clear statement of research objectives or research question: Original sentence: "The purpose of this manuscript is to report on data and feasibility from the rehabilitation and exercise Triage Clinic (TC), a component of the Alberta Cancer Exercise-Neuro Oncology study (ACE-Neuro)." Revised sentence: "This manuscript aims to present data and evaluate the feasibility of the rehabilitation and exercise Triage Clinic (TC) as part of the Alberta Cancer Exercise-Neuro Oncology study (ACE-Neuro)."

Clarify the significance of the research and its contribution to the field. How does the study aim to address the identified gaps or limitations?

Materials and Methods section:

The methods section requires some improvements in terms of clarity:

-            Elaboration on how feasibility was assessed quantitatively and qualitatively.             

Specify the specific measures used to calculate the referral, enrollment, and TC attendance rates. For example, clarify the time frame over which these rates were calculated. In addition, mention any specific criteria or thresholds for determining the feasibility rates (e.g., why a referral rate of at least 50% was chosen).

-            For the qualitative assessment of feasibility:

Provide more information on the semi-structured interviews, such as the number of participants interviewed and any specific questions or prompts (mention the data saturation).

The results are presented clearly and organized, with tables and figures to provide a visual representation.

Overall, the presentation of the results seems satisfactory and effectively communicates the necessary information.

Discussion/conclusion

The authors should provide more specific details or statistical evidence to strengthen their claims. For example, instead of stating that "the TC was safe," authors can include specific safety data, such as the absence of adverse events and any relevant statistics related to participant safety. If the data were from qualitative interviews, put a more explicit statement.

The authors can further elaborate on how the results specifically contribute to the feasibility and acceptance of the TC.

The authors can consider including more specific data or examples to support their claim regarding the lack of standardized identification of patients with functional impairment. They can provide information on existing screening practices or studies that have assessed the prevalence of functional impairments among cancer patients.

The authors can mention screening tools or practices currently used in cancer care settings and compare them to the proposed CREST tool. In addition, I believe authors can provide statistics or references to studies highlighting the prevalence of functional impairments among cancer patients, emphasizing the need for standardized identification methods.

Author Response

Dear Reviewer 1:

Thank you for your thoughtful edits and suggestions. We have taken all comments and suggestions into consideration and have made the following edits:

  1. The abstract concisely summarizes the study, outlining the purpose, methodology, and critical findings. It offers insights into the feasibility and effectiveness of the TC in facilitating rehabilitation and exercise referrals for patients with brain tumors. The mention of the proposed screening tool indicates a potential strategy for improving the identification of patients needing rehabilitation and exercise interventions.
    1. Few areas that could be improved:
    2. Background and context: The abstract could provide more background information to contextualize the study. It would be beneficial to briefly discuss the existing literature on physical impairment and sedentary behavior in neuro-oncology populations.
      1. Thank you for these suggestions. Overall, we were limited in the abstract by the 200 word count, but have attempted to make edits as per your suggestions. We have specified functional impairments that specifically affect neuro-oncology patients, including balance, motor skills, and sedentary behaviour. To facilitate these changes, our word count is now at 228, which we hope will be acceptable by the editors.
    3. Referral rates explanation: The abstract mentions that referral rates into the Triage Clinic (TC) were not met, but it does not explain this outcome clearly. Providing some insights into the potential reasons for the lower-than-expected referral rates would enhance the understanding of the study's limitations.
      1. Again, we are limited in the abstract by the word count of 200, but have included reasons why referral rates were potentially not met. This includes clinic staff forgetting to refer patients, and uncertainty of who was appropriate for rehabilitation.
    4. Impact and implications: The abstract could expand on the implications and potential impact of the study. It would be helpful to discuss how the findings of the feasibility assessment and the proposed screening tool (CREST) could influence clinical practice and improve patient outcomes.
      1. Thank you for this suggestion. We have added a sentence at the end of the abstract to suggest how the CREST could impact clinical practice and improve patient experience.
    5. Introduction:
      1. The introduction provides sufficient background information and includes relevant references.
      2. The introduction can be improved by reorganizing the content for better flow and coherence. In addition, the information is currently presented in a somewhat fragmented manner.
        1. Thank you for this feedback. We have re-arranged the introduction to include the evidence behind the ‘evidence to practice’ gap in screening, triage and rehabilitation services up front, and the reason for studying neuro-oncology patients further down. We hope this helps to improve the flow and coherence.
      3. Some sentences could be rephrased for clarity and readability.
        1. Thank you for this suggestion. We have re-organized according to your recommendations below.
      4. It would be helpful to provide a clear statement of the research objectives or research question early in the introduction to guide the reader.
        1. Thank you for this suggestion. We have clarified the study objectives and research aim, as per your suggestions below.
      5. For example:
      6. Reorganizing the content for better flow and coherence: Original sentence: "These patients often report ongoing cognitive, physical, and psychological impairment that is not adequately addressed, and are currently an understudied population in rehabilitation oncology literature [5, 6]." Revised sentence: "Neuro-oncology patients, who frequently experience cognitive, physical, and psychological impairment, often find that their needs are not adequately addressed. Surprisingly, this patient population remains understudied in rehabilitation oncology [5, 6]."
        1. Thank you for this great suggestion. We have implemented this change and agree it improves readability.
      7. Rephrasing for clarity and readability: Original sentence: "Once patients are screened and identified as having functional deficits or concerns, triage and referral pathways are necessary." Revised sentence: "Following the identification of patients with functional deficits or concerns through screening, it becomes essential to establish triage and referral pathways."
        1. This change has been made to improve readability.
      8. Providing a clear statement of research objectives or research question: Original sentence: "The purpose of this manuscript is to report on data and feasibility from the rehabilitation and exercise Triage Clinic (TC), a component of the Alberta Cancer Exercise-Neuro Oncology study (ACE-Neuro)." Revised sentence: "This manuscript aims to present data and evaluate the feasibility of the rehabilitation and exercise Triage Clinic (TC) as part of the Alberta Cancer Exercise-Neuro Oncology study (ACE-Neuro)."
        1. Thank you again for the suggestion. We have made the edits as per your recommendation and agree it reads better.
      9. Clarify the significance of the research and its contribution to the field. How does the study aim to address the identified gaps or limitations?
        1. We have expanded the introduction to include a paragraph on the purpose of this work and how it will contribute to the current gap in care. We hope this helps to clarify the significance of this work.

MATERIALS AND METHODS:

The methods section requires some improvements in terms of clarity:

  1. Elaboration on how feasibility was assessed quantitatively and qualitatively. Specify the specific measures used to calculate the referral, enrolment, and TC attendance rates. For example, clarify the time frame over which these rates were calculated. In addition, mention any specific criteria or thresholds for determining the feasibility rates (e.g., why a referral rate of at least 50% was chosen).
    1. Thank you for this comment. We have included a further description of how we determined our feasibility criteria to the methods section.
  2. For the qualitative assessment of feasibility: Provide more information on the semi-structured interviews, such as the number of participants interviewed and any specific questions or prompts (mention the data saturation).
    1. Thank you for your suggestions. We did include the number of participants interviewed in our results section, but to add clarity to the methods and address the mention of data saturation, we added that participants were recruited via purposive sampling for the interviews. In addition, we added examples of questions asked during the interviews. These can be found in Appendix C and are mentioned in the methods section.

RESULTS:

The results are presented clearly and organized, with tables and figures to provide a visual representation.

Overall, the presentation of the results seems satisfactory and effectively communicates the necessary information.

DISCUSSION AND CONCLUSIONS:

  1. The authors should provide more specific details or statistical evidence to strengthen their claims. For example, instead of stating that "the TC was safe," authors can include specific safety data, such as the absence of adverse events and any relevant statistics related to participant safety. If the data were from qualitative interviews, put a more explicit statement.
    1. Thank you for this comment. We have added your suggestion, including reporting the absence of adverse events, and more specific statements from the qualitative data regarding safety.
  2. The authors can further elaborate on how the results specifically contribute to the feasibility and acceptance of the TC.
    1. We have tried to be more specific throughout the discussion when referring to the feasibility and acceptability of the TC (referencing both the quantitative and qualitative data). In addition, we have rearranged the discussion to better address feasibility up front, and then further expand on the low referral rate and solution with the CREST tool later in the discussion. We hope this addresses your comment.
  3. The authors can consider including more specific data or examples to support their claim regarding the lack of standardized identification of patients with functional impairment. They can provide information on existing screening practices or studies that have assessed the prevalence of functional impairments among cancer patients.
    1. Thank you for this suggestion. We have added more examples and literature to support the high rate of physical impairment among cancer patients. We also comment on the lack of screening as a contributor to high impairment rates, as many patient concerns go unidentified and therefore unaddressed.
  4. The authors can mention screening tools or practices currently used in cancer care settings and compare them to the proposed CREST tool. In addition, I believe authors can provide statistics or references to studies highlighting the prevalence of functional impairments among cancer patients, emphasizing the need for standardized identification methods.
    1. Thank you. Along with the previous comment, we have added references to show the high prevalence of physical impairment among cancer survivors, and the lack of screening as a key contributor to this high rate of unaddressed physical impairment. In addition, we have added a sentence about the lack of any other rehabilitation screening tool (to the best of our knowledge) and further description of how existing tools were designed for research purposes, not screening purposes, and are not specific to the cancer population. We hope this helps to further clarify the discussion.

Reviewer 2 Report

The authors have assessed brain tumor patients' rehabilitation and exercise needs and tested the feasibility of a Rehabilitation Triage Clinic (TC) for referral to resources. The TC was safe and facilitated referrals, but referral rates were not met. Lack of oncology-clinic screening may hinder the identification of sedentary or impaired patients. They proposed the Cancer Rehabilitation & Exercise Screening Tool (CREST) in their updated model. The contributions of the work are significant and have benefits in understanding the rehabilitation and exercise needs of patients after cancer. The manuscript could be accepted with a few minor revisions:

1. The authors are suggested to include more details in Section 2.7, statistical analysis (qualitative data).

2. On page 14, the authors are suggested to provide the reasoning/justification of different outcomes clearly. For example, in the observation, "Despite this scoring, participants were still able to perform 480 basic exercises, making them appropriate for referral to the ACE-Neuro exercise study, " what is the significance of such scoring?

3. To improve the paper's readability, The authors should mention the difference between their previous study [5] and the present study in the Introduction section. 

4. The reference [5] and [37] are the same (authors' paper) and need to be corrected.

Author Response

Dear Reviewer 2:

Thank you for your thoughtful edits and suggestions.

  1. The authors are suggested to include more details in Section 2.7, statistical analysis (qualitative data).
    1. Thank you for this suggestion. We added an additional line to section 2.7 for qualitative data analysis. However, we do believe this paragraph nicely captures the process for conventional content analysis and thus have kept it succinct.
  2. On page 14, the authors are suggested to provide the reasoning/justification of different outcomes clearly. For example, in the observation, "Despite this scoring, participants were still able to perform 480 basic exercises, making them appropriate for referral to the ACE-Neuro exercise study, " what is the significance of such scoring?
    1. Thank you for this suggestion. We have added additional references to describe the cut off scores on the Short Physical Performance Battery, and have further explained why we changed the initial cut off of 10/12 to 5/12. We hope this makes it easier to understand the reasoning for this change in the context of participant functional abilities.
  3. To improve the paper's readability, The authors should mention the difference between their previous study [5] and the present study in the Introduction section. 
    1. Thank you for this suggestion. Reference 5 (Daun et al., 2021) refers to our study protocol paper, not previously published results. To ensure this is more clear, we have removed this reference from the early introduction literature review to improve clarity, and wait to reference this paper in the methods section when referring to the previously published methods.
  4. The reference [5] and [37] are the same (authors' paper) and need to be corrected.
    1. Thank you for identifying this error. This has now been corrected.